# Omadacycline, Eravacycline, and Tigecycline Express Anti-*Mycobacterium abscessus* Activity *In Vitro*

Anqi Li,[a,b] Siyuan He,[a,b] Jingren Li,[a,b] Zhemin Zhang,[a,b] Bing Li,[a,b] Haiqing Chu[a,b,c]

[a]Department of Respiratory and Critical Care Medicine, Shanghai Pulmonary Hospital, School of Medicine, Tongji University, Shanghai, China
[b]School of Medicine, Tongji University, Shanghai, China
[c]Shanghai Key Laboratory of Tuberculosis, Shanghai Pulmonary Hospital, School of Medicine, Tongji University, Shanghai, China

**ABSTRACT** *Mycobacterium abscessus* infections are increasing worldwide necessitating the development of new antibiotics and treatment regimens. The utility of third-generation tetracycline antibiotics was reestablished; their anti-*M. abscessus* activity needs further study. The activities of omadacycline (OMC), eravacycline (ERC), tigecycline (TGC), and sarecycline (SAC) were tested against two reference strains and 193 clinical *M. abscessus* isolates at different temperatures (30°C and 37°C). The minimum bactericidal concentrations (MBCs) of the four drugs were determined to distinguish between their bactericidal and bacteriostatic activities. The MICs of OMC, ERC, and TGC for the reference strains and clinical isolates were summarized and compared. OMC, ERC, and TGC exhibited a high level of bacteriostatic activity against *M. abscessus*. The MICs of OMC and ERC for *M. abscess* remained stable, while the MICs of TGC for the isolates/strains increased with increasing temperature. Notably, the MICs of OMC for *M. abscessus* isolates obtained in the United States are lower than for those obtained in China.

**IMPORTANCE** The antimicrobial activities of four third-generation tetracycline-class drugs, omadacycline (OMC), eravacycline (ERC), tigecycline (TGC), and sarecycline (SAC), were determined for 193 *M. abscessus* isolates. The activities of the four drugs at two different temperatures (30°C and 37°C) were also tested. OMC, ERC, and TGC exhibited significant activity against *M. abscessus*. The anti-*M. abscessus* activity of TGC increased when the temperature was increased from 30°C to 37°C; the activities of OMC and ERC, on the other hand, remained the same. We found that *in vitro* MICs of OMC against Chinese and American isolates were distinct. Evaluations in *in vivo* models of *M. abscessus* disease or in the clinical setting will provide more accurate insight into potency of OMC against distinct isolates.

**KEYWORDS** *Mycobacterium abscessus*, *in vitro*, tetracycline, omadacycline, eravacycline, tigecycline, sarecycline

Infections caused by nontuberculous mycobacteria (NTM), which include both rapidly and slowly growing organisms, are becoming more common worldwide (1). The *Mycobacterium abscessus* complex, consisting of three subspecies, *abscessus*, *massiliense*, and *bolletii*, constitutes the main pathogenic rapidly growing mycobacteria (RGM) (2, 3). In the United States, subspecies *abscessus* is primarily responsible for pulmonary infections caused by RGM; nevertheless, infections by subspecies *massiliense* are at least as common in other parts of the world (4). The increased global prevalence of *M. abscessus* poses a serious threat to human health; it is naturally multidrug resistant and difficult to treat (5). *M. abscessus* infections are associated with high rates of morbidity and mortality, particularly for patients with chronic lung diseases such as cystic fibrosis and bronchiectasis (6–8). According to ATS/ERS/ESCMID/IDSA guidelines, the multidrug treatment regimen for *M. abscessus* pulmonary disease consists of ≥3 drugs, including a macrolide if the bacteria are susceptible, and ≥4 drugs if the bacteria are resistant to macrolides (9). Usually, the regimen is composed of an oral macrolide, intravenous amikacin, and ≥1 additional antibiotics, e.g., tigecycline (TGC),

Address correspondence to Haiqing Chu, chu_haiqing@126.com, or Bing Li, libing044162@163.com.

The authors declare no conflict of interest.

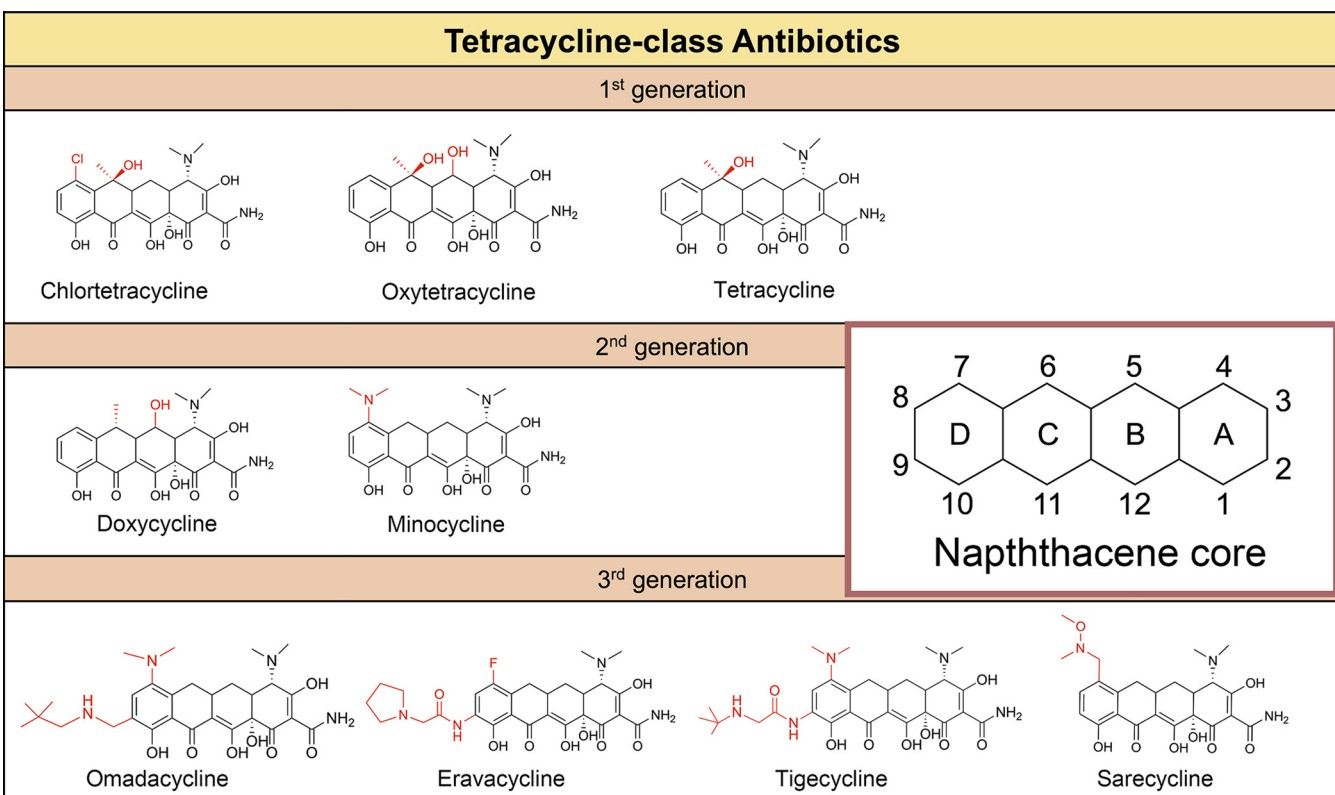

**FIG 1** Chemical structures of tetracycline-class antibiotics. First-generation tetracyclines: chlortetracycline, oxytetracycline, and tetracycline. Second-generation tetracyclines: doxycycline and minocycline. Third-generation tetracyclines: omadacycline, eravacycline, tigecycline, and sarecycline. The inset, illustrating the DCBA naphthacene core, indicates the carbon atom assignments for rings A to D.

imipenem, or cefoxitin administered intravenously (10). While treatment with TGC has achieved certain success, adverse effects such as nausea and vomiting, and the requirement for intravenous administration limit its use (11, 12). The development of new therapies that utilize antibiotics that exhibit comparable or better anti-*M. abscessus* activity, improved bioavailability, and fewer adverse side effects are urgently needed.

Third-generation tetracycline-class drugs that include TGC, omadacycline (OMC), eravacycline (ERC), and sarecycline (SAC) received US Food and Drug Administration (FDA) approval in recent years. Compared to the chemical structures of first- and second-generation tetracycline-class antibiotics, third-generation antibiotics possess more complex groups on the D ring (Fig. 1). TGC and ERC are only available in IV formulations, SAC is available as an oral tablet, and OMC is available in both oral and IV formulations (13). In 2018, the FDA approved OMC for clinical use in patients with acute bacterial skin and skin structure infections, and community-acquired bacterial pneumonia; ERC and SAC were approved for treatment of complicated intraabdominal and severe acne vulgaris infections, respectively. OMC expresses anti-*M. abscessus* activity *in vitro* and *in vivo*; ERC exhibits activity *in vitro* (14–16). Notably, Brown-Elliott et al. obtained up to 15-fold differences in the calculated $MIC_{50}$ of OMC and TGC for *M. abscessus* isolates in three different studies; disparities in incubation temperatures were suggested as a possible reason (14, 16, 17). No MIC assays were conducted using the same clinical isolates in a single study. Here, the anti-*M. abscessus* activities of OMC, ERC, SAC, and TGC were tested at 30°C and 37°C *in vitro*, and the results were compared to those reported in published studies.

## RESULTS

**OMC, ERC, and TGC are active against *M. abscessus*.** Two *M. abscessus* reference strains (ATCC 19977 and CIP 108297) and 193 clinical isolates were collected for antibiotic susceptibility testing. Detailed test results for the activity of four tetracycline analogs (OMC, ERC, TGC and SAC) against 193 clinical isolates cultured at 30°C are presented in Table 1

**TABLE 1** The MICs of omadacycline, eravacycline, tigecycline, and sarecycline for 193 clinical *M. abscessus* isolates cultured at 30°C[a]

| Antimicrobial agent[b] | Subspecies (n = 193) | MIC$_{50}$[c] (µg/mL) | MIC$_{90}$[c] (µg/mL) | MIC range (µg/mL) | No. (%) of isolates sensitive to antimicrobial agent concentrations (µg/mL) | | | | | | | | |
|---|---|---|---|---|---|---|---|---|---|---|---|---|---|
| | | | | | 0.063 | 0.125 | 0.25 | 0.5 | 1 | 2 | 4 | 8 | >8 |
| OMC | *abscessus* (n = 147) | 1 | 4 | 0.25–4 | 0 | 0 | 2 (1.3) | 30 (13.6) | 52 (35.3) | 58 (39.4) | 15 (10.2) | 0 | 0 |
| | *massiliense* (n = 46) | 2 | 4 | 0.5–8 | 0 | 0 | 0 | 5 (10.8) | 15 (32.6) | 14 (30.4) | 10 (21.7) | 2 (4.3) | 0 |
| ERC | *abscessus* (n = 147) | 1 | 4 | 0.063–4 | 1 (0.6) | 6 (4.0) | 17 (11.5) | 45 (30.6) | 38 (25.8) | 24 (16.3) | 16 (10.8) | 0 | 0 |
| | *massiliense* (n = 46) | 1 | 4 | 0.125–8 | 0 | 2 (4.3) | 9 (19.5) | 6 (13) | 11 (23.9) | 10 (21.7) | 5 (10.8) | 3 (6.5) | 0 |
| TGC | *abscessus* (n = 147) | 0.5 | 2 | 0.063–4 | 1 (0.6) | 6 (4) | 30 (20.4) | 58 (39.4) | 32 (21.7) | 18 (12.2) | 2 (1.3) | 0 | 0 |
| | *massiliense* (n = 46) | 1 | 2 | 0.125–4 | 0 | 2 (4.3) | 12 (26) | 6 (13) | 16 (34.7) | 7 (15.2) | 3 (6.5) | 0 | 0 |
| SAC | *abscessus* (n = 147) | >8 | >8 | 4– >8 | 0 | 0 | 0 | 0 | 0 | 0 | 3 (2) | 0 | 145 (98) |
| | *massiliense* (n = 46) | >8 | >8 | 8– >8 | 0 | 0 | 0 | 0 | 0 | 0 | 0 | 1 (2.1) | 45 (97.9) |

[a]MIC values were determined based upon the CLSI document M24-A2 guidelines for aerobic bacteria.

[b]OMC, omadacycline; ERC, eravacycline; TGC, tigecycline; SAC, sarecycline.

[c]MIC$_{50}$ and MIC$_{90}$ are defined as the concentrations at which 50% and 90% of the clinical isolates tested, respectively, were inhibited.

and Table S1. At 30℃ (CLSI recommended temperature), the MICs of OMC ranged from 0.25 to 8 $\mu$g/mL: the $MIC_{50}$ was 1 $\mu$g/mL and the $MIC_{90}$ was 4 $\mu$g/mL for subsp. *abscessus*; the $MIC_{50}$ was 2 $\mu$g/mL and the $MIC_{90}$ was 4 $\mu$g/mL for *massiliense*. The MICs of ERC ranged from 0.063 to 8 $\mu$g/mL: the $MIC_{50}$ was 1 $\mu$g/mL and the $MIC_{90}$ was 4 $\mu$g/mL for both subspecies *abscessus* and *massiliense*. The MICs of TGC ranged from 0.063 to 4 $\mu$g/mL: the $MIC_{50}$ was 0.5 $\mu$g/mL and the $MIC_{90}$ was 2 $\mu$g/mL for subsp. *abscessus*; the $MIC_{50}$ was 1 $\mu$g/mL and the $MIC_{90}$ was 2 $\mu$g/mL for subsp. *massiliense*. The MICs of SAC ranged from 4 to >8 $\mu$g/mL; both the $MIC_{50}$ and $MIC_{90}$ were >8 $\mu$g/mL, exhibiting significantly less activity than OMC, ERC, and TGC toward *M. abscessus* isolates. To summarize, TGC, ERC, and OMC were active against *M. abscessus*. TGC exhibited the greatest activity; OMC and ERC also expressed high anti-*M. abscessus*; SAC was inactive.

**OMC, ERC, and TGC exhibit bacteriostatic activity.** The MBCs of OMC, ERC, and TGC were tested on the two reference strains and 18 randomly selected isolates following MIC testing, and the MBC/MIC ratios were determined (see Table S2 in the supplemental material). SAC was excluded since the MICs were >8 $\mu$g/mL. OMC, ERC, and TGC exhibited bacteriostatic activity with MBC/MIC ratios >4 or >8 against all strains tested at both 30℃ and 37℃.

**Temperature impacts the MIC of TGC, but not of OMC or ERC for *M. abscessus*.** The MICs of the 4 tetracycline analogs for 29 clinical *M. abscessus* isolates selected at random from 193 isolates were determined after 3 days incubation at 30℃ and 37℃. A detailed comparison of the MICs for the 29 strains is shown in Table S3. Paired Wilcoxon tests were performed. SAC was excluded from statistical analysis since all the MICs were >8 $\mu$g/mL. The statistical results and MIC distribution are shown in Fig. 2. The MICs of TGC for the *M. abscessus* isolates were higher when the incubation temperature was 37℃, compared to 30℃ ($P < 0.05$); increasing the incubation temperature to 37℃, however, did not impact the MICs of OMC or ERC.

**Comparison of the MICs of OMC, ERC, TGC, and SAC for *M. abscessus* determined in different studies.** The MICs of the 4 tetracycline analogs for clinical *M. abscessus* isolates obtained in the current study and in 5 studies reported previously were collected and compared (Table 2) (14, 15, 17–19). The MICs for the *M. abscessus* reference strains were also compared (Table S4). OMC, ERC, and TGC were active toward the *M. abscessus* isolates collected in all the studies: the $MIC_{50}$ of OMC ranged from 0.12 to 2 $\mu$g/mL, and the $MIC_{90}$ ranged from 0.25 to 8 $\mu$g/mL; the $MIC_{50}$ of ERC ranged from 0.12 to 1 $\mu$g/mL, and the $MIC_{90}$ ranged from 0.25 to 8 $\mu$g/mL; the $MIC_{50}$ of TGC ranged from 0.12 to 1 $\mu$g/mL, and $MIC_{90}$ ranged from 0.25 to 8 $\mu$g/mL. Kaushik et al. (14) reported the same $MIC_{90}$ of OMC (4 $\mu$g/mL) and TGC (2 $\mu$g/mL) for subsp. *abscessus*, which were higher than the rest. Zhang et al. and Chew et al. reported the same $MIC_{50}$ and $MIC_{90}$ for ERC and TGC, which were the lowest among all the studies (18, 19). Remarkably, the MICs of OMC, ERC, and TGC determined in the present study were higher than those reported in studies published previously, and the differences remained after considering the effect of temperature. Among the 6 studies referenced in Table 2, 3 were conducted in the USA, 2 in China, and 1 in Singapore. In general, the MICs of OMC determined in the USA studies were lower than those determined in China (14, 15, 17, 19). Differences in the MICs determined for the reference strains were also obvious (Table S4). Notably, Zhang et al. (19) reported that both the $MIC_{50}$ and $MIC_{90}$ of SAC were >8 $\mu$g/mL, consistent with the results presented in the current study and supporting the poor anti-*M. abscessus* activity of SAC.

## DISCUSSION

*M. abscessus* is the second most common nontuberculous *mycobacterium* that causes pulmonary diseases. It is an opportunistic pathogen, which is also involved in systemic and disseminated infections; multidrug resistance limits the therapeutic options (1, 5, 20). Multidrug therapy is recommended for treating *M. abscessus* infections, consisting of at least 3 drugs, including macrolides. Since drug resistance is prevalent, new antibiotics and regimens are needed. Tetracycline-class drugs demonstrate a broad spectrum of activity and have been used extensively to treat both humans and animals (21). However, the growing rates of resistance and occurrence of adverse events limit the utility of first-generation tetracyclines

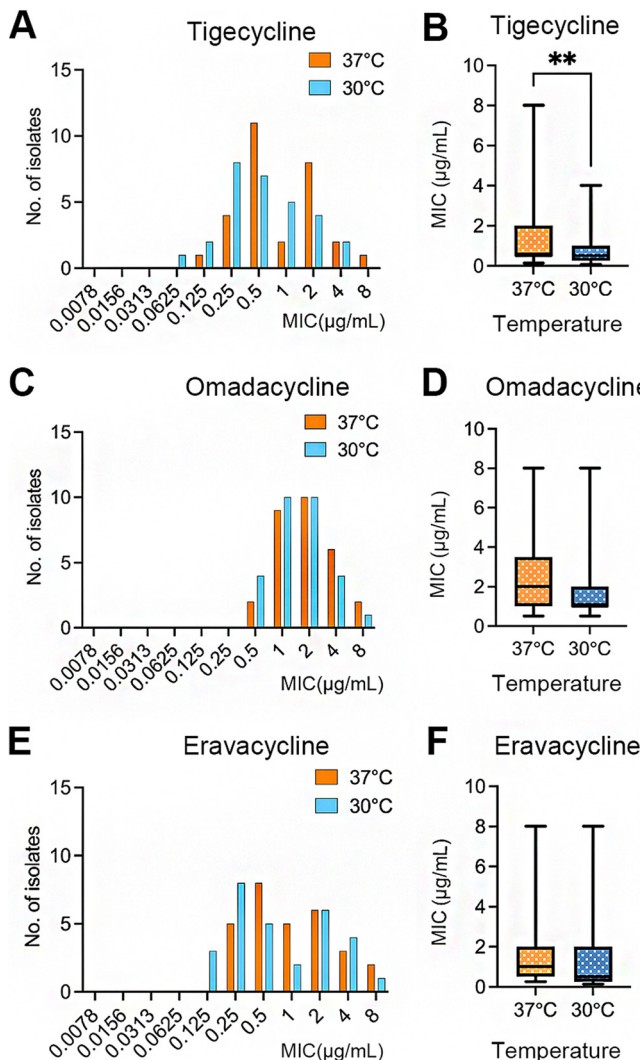

**FIG 2** Distribution and paired statistical analysis of the MICs of tigecycline, omadacycline, and eravacycline for 29 clinical *M. abscessus* isolates incubated at 30°C and 37°C. (A, C, E) Distribution of the MICs of tigecycline (A), omadacycline (C) and eravacycline (E). (B, D, F) Paired statistical analyses of the MICs of tigecycline (B), omadacycline (D) and eravacycline (F). Significantly different: **, $P < 0.01$.

and second-generation minocycline and doxycycline. TGC was the first third-generation antibiotic included in the treatment of *M. abscessus*. Novel third-generation tetracycline-class antibiotics, i.e., OMC, ERC, and SAC, were approved for clinical use by the US FDA in 2018. Their anti-*M. abscessus* activities require further delineation. In the present study, the anti-*M. abscessus* activities of OMC, ERC, TGC, and SAC against 193 clinical isolates obtained at Shanghai Pulmonary Hospital were determined. OMC, ERC, and TGC were active against *M. abscessus* reference strains and clinical isolates; SAC was inactive.

The effect of incubation temperature on the anti-*M. abscessus* activities expressed by OMC, ERC, and TGC and the calculation of the MICs for the same *M. abscessus* isolates remained to be determined (17). According to CLSI document M24-A2, antimycobacterial susceptibility testing of rapidly growing mycobacteria should be conducted at an incubation temperature between 28 and 30°C, and determined after 72 h. Previous susceptibility tests were conducted at either 30°C or 37°C. Analysis of OMC, ERC, and TGC in the current study revealed that increasing the incubation temperature from 30°C to 37°C *in vitro* decreased the activity and increased the MIC of TGC that affected 29 randomly selected *M. abscessus* isolates. Previously, Ng et al. (22) reported similar results, i.e., increasing the temperature from 37°C to 42°C induced the transient resistance of the *M. abscessus*

**TABLE 2** Comparison of the MICs of omadacycline, eravacycline, tigecycline, and sarecycline obtained for *M. abscessus* isolates in 6 studies[a]

| Study (yr and region) | n | Incubation temp/time | Subspecies | MIC$_{50}$ ($\mu$g/mL) | | | | MIC$_{90}$ ($\mu$g/mL) | | | |
|---|---|---|---|---|---|---|---|---|---|---|---|
| | | | | OMC | ERC | TGC | SAC | OMC | ERC | TGC | SAC |
| Kaushik et al. (14) (2019, USA) | 16 | 30°C/3 d | *abscessus* | 2 | 0.5 | 1 | | 4 | 2 | 2 | |
| | 12 | 30°C/3 d | *massiliense* and *bolletii* | 1 | 0.25 | 1 | | 2 | 0.5 | 2 | |
| Brown-Elliott et al. (17) (2020, USA) | 20 | 30°C/3 d | *abscessus* | 0.12[b] | | 0.12 | | 0.25[b] | | 0.25 | |
| | 3 | 30°C/3 d | *massiliense* | 0.12[b] | | 0.25 | | | | | |
| Chew et al. (18) (2020, Singapore) | 218 | 30°C/3−5 d | ND | | 0.12 | 0.5 | | - | 0.25 | 1 | |
| Nicklas et al. (15) (2022, USA) | 12 | 30°C/3 d | *abscessus* | 0.25 | | 0.188 | | 0.5 | | 0.25 | |
| | 9 | 30°C/3 d | *massiliense* | 0.375 | | 0.188 | | 1 | | 0.5 | |
| | 10 | 30°C/3 d | ND | 0.125 | | 0.125 | | 0.5 | | 0.25 | |
| Zhang et al. (19) (2022, China) | 44 | 37°C/3 d | *abscessus* | 0.5 | 0.12 | 0.5 | >8 | 1 | 0.25 | 1 | >8 |
| | 29 | 37°C/3 d | *massiliense* | 1 | 0.12 | 0.5 | >8 | 2 | 0.25 | 1 | >8 |
| Current study (2023, China) | 147 | 30°C/3 d | *abscessus* | 1 | 1 | 0.5 | >8 | 4 | 4 | 2 | >8 |
| | 46 | 30°C/3 d | *massiliense* | 2 | 1 | 1 | >8 | 4 | 4 | 2 | >8 |
| | 23 | 37°C/3 d | *abscessus* | 2 | 0.5 | 0.5 | >8 | 4 | 4 | 2 | >8 |
| | 6 | 37°C/3 d | *massiliense* | 2 | 1 | 1 | >8 | 8 | 8 | 8 | >8 |

[a]*n*, number of isolates; ND, not determined; OMC, omadacycline; ERC, eravacycline; TGC, tigecycline; SAC, sarecycline.
[b]Concentrations that inhibit 100% of growth.

reference strain ATCC 19977 to TGC. Temperature, however, did not affect the MICs of OMC and ERC, rather, they appeared to demonstrate greater stability.

A review of the current and published studies reveals several points of comparison. Here, OMC expressed the same or slightly less activity than either ERC or TGC. Similarly, Zhang et al. reported that OMC exhibited a higher MIC for *M. abscessus* than ERC *in vitro* and concluded that degradation of OMC was a contributing factor (19). OMC decay is evident. The concentration of OMC in a solution composed of <0.01% dimethyl sulfoxide (DMSO) in sterile water decreased ~50% after 24 h incubation at 37°C in one study (23). In a second study, OMC decreased to 16.7 to 27.1% of the initial concentration during a 5-day incubation period at 37°C (19). Additionally, ERC exhibited the same or a smaller MIC than TCG for *M. abscessus* in 3 previous studies (14, 18, 19). In contrast, the MIC measured for ERC in the current study was the same or slightly greater than that measured for TGC. To ensure accuracy, the MIC of ERC for 2 reference strains and 20 isolates chosen at random was determined in assays repeated 4 to 5 times. The difference in measured MICs for ERC could reflect the fact that one study was conducted in the USA while two were conducted in China; variation in the number of multidrug-resistant isolates included in studies conducted at different sites could also be a contributing factor. Indeed, 99% of all isolates in the current study were resistant to multiple drugs. Similarly, Kaushik et al. (14) reported that the isolates in their study were resistant to nearly all antibiotics used to treat *M. abscessus* infections. More research involving additional *M. abscessus* isolates obtained in multiple centers is required to confirm the activity of ERC. Finally, the MICs of OMC for *M. abscessus* determined by the two studies reported by Brown-Elliott and by Nicklas, and conducted in the USA were lower than those determined by studies conducted in China, reported by Zhang et al. and herein (15, 17, 19). Presumably, the Chinese isolates are more resistant to OMC than those recovered in the USA; additional epidemiological investigations are needed.

The MICs of OMC, ERC, and TGC determined for the *M. abscessus* reference strains vary significantly between studies (Table S4). Bax et al. (24) reported, for example, that the MIC of both OMC and TGC for *M. abscessus* CIP 104536 was 4 $\mu$g/mL. This value is 15-fold higher than the value determined for *M. abscessus* ATCC 19977 in a study conducted by Zhang et al. (19, 24). Such differences undoubtedly relate to the reference stain, stability and storage of the antibiotic, and the specific experimental protocol used, e.g., time and temperature of incubation (17, 25–28).

The anti-*M. abscessus* activity exhibited by SAC differed significantly from that expressed by OMC, ERC and TGC. Differences in the chemical structures could be a contributing factor. The addition of a lipophilic side group at the C9 position on the D ring of OMC, ERC, and TGC enhances the antibiotic activity and enables the drugs to circumvent tetracycline resistance (13). SAC has no group at the C9 position. Notably, a comparison of the sequencing

data derived from all *M. abscessus* isolates failed to reveal any obvious genetic differences that could account for the reduced activity of SAC compared to the other 3 tetracycline analogs (data not shown).

TGC is an established component of multidrug regimens used to treat *M. abscessus* infections (1, 20, 29). Its clinical utility is limited, however, by the fact that only an intravenous formulation is available for administration and that adverse effects (i.e., nausea and vomiting) are reported in >90% of cases (12). ERC showed similar or better anti-*M. abscessus* activity than TGC *in vitro* although its use is limited by gastrointestinal (GI) disturbances in 62.5% of treated patients, and the availability of an intravenous formulation only (14, 19, 30). OMC exhibits activity similar to TGC, but achieved higher, sustained concentrations in plasma, epithelial lining fluid and alveolar cells in a pharmacokinetic study conducted with healthy adult subjects (31). A preliminary, multicenter study of regimens containing OMC showed clinical success in 75.0% (9/12) of *M. abscessus*-infected patients; 58.3% (7/12) of those infections were of pulmonary origin (32). Furthermore, both oral and intravenous formulations of OMC are available, and it is significantly better tolerated than either TGC or ERC. One study reported a marked difference in the incidence of nausea in the OMC-treated group (1/42, 2.4%) versus the TGC-treated group (10/21, 47.6%) (31). A study conducted in China reported only a slight incidence of GI disturbance (6/50, 12%) due to OMC treatment (32). OMC offers a promising option for inclusion in the *M. abscessus* infection treatment regimen considering its ample anti-*M. abscessus* activity expressed *in vitro*, pharmacokinetic advantages, and remarkably decreased incidence of GI disturbance.

The limitations of the current study are 3-fold. First, all the isolates were collected in a single center from patients diagnosed with NTM pulmonary infection. Second, susceptibility assays were only conducted *in vitro*; intracellular and *in vivo* comparisons are lacking. Results from *in vitro* studies do not recapitulate *in vivo* conditions and MICs from *in vitro* studies have not been consistently informative for treating *M. abscessus* disease in the clinic. Third, only the effects of incubation at 30°C and 37°C on the susceptibility of *M. abscessus* to tetracycline analogs were tested; a broader temperature range should be examined.

In summary, OMC, ERC, and TGC exhibited bacteriostatic activity assessed *in vitro* toward a large panel of clinical *M. abscessus* isolates. Elevating the temperature at which anti-*M. abscessus* activity was assessed increased the MIC of TGC, but not of ERC or OMC. Comparing the MICs determined in different studies consistently demonstrated that OMC, ERC, and TGC were active against all *M. abscessus* isolates, SAC was not. OMC showed great potential for incorporation into *M. abscessus* treatment regimens. It was the only oral third-generation tetracycline drug characterized by preliminary clinical effectiveness, pharmacokinetic advantages, and fewer GI disturbances.

## MATERIALS AND METHODS

**Bacterial strains.** Two reference strains were used: *M. abscessus* subsp. *abscessus* (ATCC 19977), purchased from the American Type Culture Collection (Manassas, VA, USA), and *M. abscessus* subsp. *massiliense* (CIP 108297), purchased from the Biological Resource Center of the Institute Pasteur (Paris, France). One hundred ninety-three clinical isolates were obtained from sputum and bronchoalveolar lavage fluid specimens of patients diagnosed with NTM lung infections in the Shanghai Pulmonary Hospital. All genome sequences were published and are available at DDBJ/ENA/GenBank (BioProject PRJNA488058, PRJNA448987, and PRJNA398137). All strains were grown at 37°C in Middlebrook 7H9 broth supplemented with 10% oleic acid-albumin-dextrose-catalase (OADC) or on Middlebrook 7H10 agar plates supplemented with 10% OADC.

**Antimicrobial agents.** OMC, ERC, TGC, and SAC (>98% purity) were purchased from MedChemExpress (Monmouth Junction, NJ, USA), solubilized in DMSO, aliquoted, stored at −80°C, and serially diluted just prior to each experiment.

**MIC determination.** Antimicrobial susceptibility testing was performed on the 2 *M. abscessus* reference strains and 193 clinical *M. abscessus* isolates using the broth microdilution method according to Clinical and Laboratory Standards Institute (CLSI) document M24-A2 (33). Briefly, bacteria in logarithmic phase of growth were diluted to 0.5 McFarland turbidity standard in sterile water then adjusted to 1 to $5 \times 10^5$ CFU/mL in cation-adjusted Mueller-Hinton broth (CAMHB). A 100-$\mu$L suspension of bacteria and 100 $\mu$L of antibiotic serially diluted 2-fold in sterile saline were suspended evenly in each well of 96-well microtiter plates; control wells contained 100 $\mu$L of suspended bacteria and 100 $\mu$L of CAMHB only. The final working concentration of the four antibiotics ranged from 0.008 to 8.0 $\mu$g/mL. Plates were sealed and incubated at 30°C or 37°C for 3 days. MICs were defined as the minimum concentration at which no visible bacterial growth was observed.

**MBC determination.** The MBC was determined following completion of MIC testing. The contents of each microtiter well containing drug concentrations greater than the MIC were suspended, and 100-$\mu$L aliquots were spread onto Middlebrook 7H10 agar plates supplemented with 0.2% glycerol and OADC. The plates were incubated for 5 days at 37°C then the CFU were counted. The MBC value was defined as the minimum drug concentration that yielded 0 CFU/mL. An MBC/MIC ratio ≤4 or >4 was used to classify an antimicrobial agent as bactericidal or bacteriostatic, respectively.

**Statistical analysis.** The MICs of antibiotics determined for *M. abscessus* isolates cultured at two different temperatures were compared by the Wilcoxon rank sum tests using GraphPad Prism 8 (GraphPad Software, San Diego, CA). *P* values <0.05 were considered statistically significant.

**Data availability.** All genome sequences were published and are available at DDBJ/ENA/GenBank (BioProject PRJNA488058, PRJNA448987, and PRJNA398137).

## SUPPLEMENTAL MATERIAL

Supplemental material is available online only.
**SUPPLEMENTAL FILE 1**, PDF file, 0.5 MB.

## ACKNOWLEDGMENTS

We sincerely thank Stephen H. Gregory (Providence, Rhode Island, USA) for editing the manuscript.

This work was funded by grants provided by the National Natural Science Foundation of China (No. 81672063, 81971973, 82270005, and 81800003), Natural Science Foundation of Shanghai (No. 20ZR1447200, 21ZR1453600, and 22ZR1452000), Medical Guide Program of Shanghai Science and Technology Committee (No. 21Y11900700), Zhongnanshan Medical Foundation of Guangdong Province (No. ZNSXS-20220069), and Clinical Research Foundation of Shanghai Pulmonary Hospital (No. FKLY20009).

The authors have no conflicts of interest to declare.

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
