## [Reviewer comments · Microbiology Spectrum]

Microbiology Spectrum

Omadacycline, Eravacycline and Tigecycline Express *Anti-Mycobacterium abscessus* Activity *In vitro*

Anqi Li, Siyuan He, Jingren Li, Zhemin Zhang, Bing Li, and Haiqing Chu

Corresponding Author(s): Haiqing Chu, Shanghai Pulmonary Hospital, Tongji University School of Medicine

Review Timeline:

Submission Date:	February 16, 2023
Editorial Decision:	April 5, 2023
Revision Received:	April 16, 2023
Accepted:	April 17, 2023

Editor: Gyanu Lamichhane

Reviewer(s): Disclosure of reviewer identity is with reference to reviewer comments included in decision letter(s). The following individuals involved in review of your submission have agreed to reveal their identity: Michael Cynamon (Reviewer #1); Ramandeep Singh (Reviewer #2)

Transaction Report:

DOI: <https://doi.org/10.1128/spectrum.00718-23>

April 5, 2023

Prof. Haiqing Chu
Shanghai Pulmonary Hospital, Tongji University School of Medicine
No. 507 Zhengmin Road
Shanghai
China

Re: Spectrum00718-23 (Omadacycline, Eravacycline and Tigecycline Express Anti-*Mycobacterium abscessus* Activity *In vitro*)

Dear Prof. Haiqing Chu:

Your manuscript was reviewed by two referees with expertise in mycobacterial experimental therapeutics. Please revise your manuscript as instructed by the two reviewers.

Also, note that all assessments of anti-M. abscessus activities were performed in vitro. As results from in vitro studies do not recapitulate in vivo conditions and MICs from in vitro studies have not been consistently informative for treating M. abscessus disease in the clinic, please include this as inherent limitation of the study. Because of this, strong statements such as in line 43: 'Notably, we found OMC was less likely effective against Chinese isolates than American isolates' should not be included. This claim is vague and inaccurate as it suggests that this study investigated actual activity of OMC of these two sets of isolates in the clinical setting. I suggest that you revise the statement to something like 'We found that in vitro MIC of OMC against Chinese and American isolates were distinct. Evaluations in in vivo models of M. abscessus disease or in the clinical setting will provide more accurate insight into potency of OMC against distinct isolates.'

When submitting the revised version of your paper, please provide (1) point-by-point responses to the issues raised by the reviewers and me as file type "Response to Reviewers," not in your cover letter, and (2) a PDF file that indicates the changes from the original submission (by highlighting or underlining the changes) as file type "Marked Up Manuscript - For Review Only". Please use this link to submit your revised manuscript - we strongly recommend that you submit your paper within the next 60 days or reach out to me. Detailed instructions on submitting your revised paper are below.

Link Not Available

Sincerely,

Gyanu Lamichhane

Journals Department
Reviewer comments:

Reviewer #1 (Comments for the Author):

The authors evaluated the in vitro activities of several third generation tetracyclines against *M. abscessus*. They found omadacycline, eravacycline, and tigecycline to have good in vitro activities against the panel of isolates they tested. These compounds did not have good bactericidal activity.

Specific comments:

lines 56-57: This statement needs to be redone too clarify their meaning.

line 68: "improved or good" would be better than great oral

line 113: Why was Middlebrook 7H10 agar used rather CAMH agar?

line 116: It would be useful for authors to define MBC in terms of the log killing required. I have usually considered a 2 log reduction to be necessary for bactericidal activity. It seems that the authors are using a more stringent definition likely 3 log reduction.

line 165: higher would be better than larger

line 166: lowest would be better than smallest

line 167: higher would be better than larger

line 171: lower would be better than smaller

line 173: consistent would be preferable to correlating

line 174: supporting would be preferable to evidencing

line 180: Combination or multidrug would be better than phase

line 181: delete "that"

line 208: higher would be preferable to larger

line 216: measured would be better than "calculated"

line 217: delete assays - "at random was determined"

line 218: measured would be preferable to "calculated"

line 226: lower would be better than "less"

line 256: remarkably would be preferable to remarkable

line 256: delete "separated"

Reviewer #2 (Comments for the Author):

1. Abstract: *M. abscessus* - It should be *Mycobacterium abscessus* in the first mention.
2. Line 106: How were MIC values determined? If only visible growth was measured, how did the authors calculate 50% and 90% inhibition (as mentioned in Table 1)? The authors should clarify this.
3. The authors should also evaluate the activity of these drugs against strains in macrophages or mice.
4. The legend to figure 1 needs to be more elaborated. The x- and y- axis of the MIC graphs need to be uniform.
5. The authors should include a figure showing structures of drugs. A figure showing structure of 1st, 2nd and 3rd generation tetracyclines will be more informative.
6. There were few grammatical errors in the manuscript. The authors should check them thoroughly.
7. The authors should also discuss the plausible reasons for differential activity of these drugs against various clinical strains. Is there any correlation between sequencing data and differential susceptibility of these strains.

Staff Comments:

Preparing Revision Guidelines

Please return the manuscript within 60 days; if you cannot complete the modification within this time period, please contact me. If you do not wish to modify the manuscript and prefer to submit it to another journal, please notify me of your decision immediately so that the manuscript may be formally withdrawn from consideration by Microbiology Spectrum.

The authors have determined the antimycobacterial activity of third generation tetracyclines; omadacycline, ervacycline, tigecycline and sarecycline against 2 reference strains and 193 clinical isolates of *M. abscessus* with incubation at two different temperatures (30°C and 37°C). They have also reviewed and compared their activity with the activity reported in 5 previous studies. The authors also show that omadacycline is also less effective against Chinese isolates in comparison to American isolates. The study lacks novelty as similar studies have been performed earlier. The present study is about evaluating antimycobacterial activity of third generation tetracyclines and is more suited for publication in another Journal. The authors may find the following comments useful to improve upon their manuscript.

1. Abstract: *M. abscessus* – It should be *Mycobacterium abscessus* in the first mention.
2. Line 106: How were MIC values determined? If only visible growth was measured, how did the authors calculate 50% and 90% inhibition (as mentioned in Table 1)? The authors should clarify this.
3. The authors should also evaluate the activity of these drugs against strains in macrophages or mice.
4. The legend to figure 1 needs to be more elaborated. The x- and y- axis of the MIC graphs need to be uniform.
5. The authors should include a figure showing structures of drugs. A figure showing structure of 1st, 2nd and 3rd generation tetracyclines will be more informative.
6. There were few grammatical errors in the manuscript. The authors should check them thoroughly.
7. The authors should also discuss the plausible reasons for differential activity of these drugs against various clinical strains. Is there any correlation between sequencing data and differential susceptibility of these strains.

31 abcess

76 OMC expresses

Fig. 1: showing structures

M. abcessus – US isolate lower MIC value in comparison to China.

Editor comments:

Your manuscript was reviewed by two referees with expertise in mycobacterial experimental therapeutics. Please revise your manuscript as instructed by the two reviewers.

Also, note that all assessments of anti-*M. abscessus* activities were performed *in vitro*. As results from *in vitro* studies do not recapitulate *in vivo* conditions and MICs from *in vitro* studies have not been consistently informative for treating *M. abscessus* disease in the clinic, please include this as inherent limitation of the study. Because of this, strong statements such as in line 43: 'Notably, we found OMC was less likely effective against Chinese isolates than American isolates' should not be included. This claim is vague and inaccurate as it suggests that this study investigated actual activity of OMC of these two sets of isolates in the clinical setting. I suggest that you revise the statement to something like 'We found that *in vitro* MIC of OMC against Chinese and American isolates were distinct. Evaluations in *in vivo* models of *M. abscessus* disease or in the clinical setting will provide more accurate insight into potency of OMC against distinct isolates.'

Response: Issues raised in the Reviewers' comments have been carefully addressed point by point. The limitations of only testing drug activity in vitro were addressed in the discussion (revised manuscript, lines 268-270). Previous strong statements were moderated according to the Editor's suggestion (Revised manuscript: lines 43-46).

Reviewer #1 (Comments for the Author):

1. lines 56-57: This statement needs to be redone too clarify their meaning.

Response: The statement was revised (revised manuscript, lines 57-59).

2. line 68: "improved or good" would be better than great oral

Response: "improved" was substituted from great (revised manuscript, line 69).

3. line 113: Why was Middlebrook 7H10 agar used rather CAMH agar?

Response: Middlebrook 7H10 agar is widely used for mycobacteria isolation, cultivation and sensitivity testing. It is also suggested as solid media for mycobacteria colony counting in CLSI standards. OADC enriched Middlebrook 7H10 agar was used for M. abscessus isolation and CFUs count for MBC determination in this study.

4. line 116: It would be useful for authors to define MBC in terms of the log killing required. I have usually considered a 2 log reduction to be necessary for bactericidal activity. It seems that the authors are using a more stringent definition likely 3 log reduction.

Response: A 3 log₁₀ reduction was used as the criterion to judge the bactericidal activity of tetracyclines in the experiments reported here in accordance with

references: Microbiology Spectrum. 2022;11(1): e0323822 and Microbiology Spectrum. 2022;10(6):e0137222.

5. line 165: higher would be better than larger

Response: "higher" was substituted for larger (revised manuscript, line 166).

6. line 166: lowest would be better than smallest

Response: "lowest" was substituted for smallest (revised manuscript, line 168).

7. line 167: higher would be better than larger

Response: higher" was substituted for larger (revised manuscript, line 169).

8. line 171: lower would be better than smaller

Response: "lower" was substituted for smaller (revised manuscript, line 172).

9. line 173: consistent would be preferable to correlating

Response: "consistent" was substituted for correlating (revised manuscript, line 175).

10. line 174: supporting would be preferable to evidencing

Response: "supporting" was substituted for evidencing (revised manuscript, line 176).

11. line 180: Combination or multidrug would be better than phase

Response: "Multidrug" was used (revised manuscript, line 181).

12. line 181: delete "that"

Response: "that" was deleted (revised manuscript, line 182).

13. line 208: higher would be preferable to larger

Response: "higher" was substituted for larger (revised manuscript, line 209).

14. line 216: measured would be better than "calculated"

Response: "measured" was substituted for calculated (revised manuscript, lines 215-216).

15. line 217: delete assays - "at random was determined"

Response: "assays" was deleted (revised manuscript, lines 217-218).

16. line 218: measured would be preferable to "calculated"

Response: "measured" was substituted for calculated (revised manuscript, line 218).

17. line 226: lower would be better than "less"

Response: *“lower” was substituted for less (revised manuscript, line 227).*

18. line 256: remarkably would be preferable to remarkable

Response: *“remarkably” was substituted for “remarkable” (revised manuscript, lines 264).*

19. line 256: delete "separated"

Response: *“separated” was deleted (revised manuscript, line 266).*

Reviewer #2 (Comments for the Author):

1. Abstract: M. abscessus - It should be Mycobacterium abscessus in the first mention.

Response: *“Mycobacterium abscessus” was written (revised manuscript, line 21).*

2. Line 106: How were MIC values determined? If only visible growth was measured, how did the authors calculate 50% and 90% inhibition (as mentioned in Table 1)? The authors should clarify this.

Response: *MIC was defined as the minimum concentration at which no visible bacterial growth was observed. In Table 1, MIC₅₀ and MIC₉₀ are defined as the concentrations at which 50% and 90% of the clinical isolates tested, respectively, were inhibited. (revised manuscript, Table 1 footnote c).*

3. The authors should also evaluate the activity of these drugs against strains in macrophages or mice.

Response: *Granted, this study is limited by the failure to include experiments that evaluate the effects of 3rd generation tetracycline-class antibiotics on M. abscessus growing in vivo (revised manuscript, lines 266-270). Future experiments testing the effects of these antibiotics on the growth of M. abscessus in mice are envisioned.*

4. The legend to figure 1 needs to be more elaborated. The x- and y- axis of the MIC graphs need to be uniform.

Response: *The legend to Figure 1 (now Figure 2) was revised. The x- and y- axes of the MIC graphs are now standardized (revised manuscript, lines 438-442, revised Figure 2).*

5. The authors should include a figure showing structures of drugs. A figure showing structure of 1st, 2nd and 3rd generation tetracyclines will be more informative.

Response: *The revised manuscript now includes a Figure (1) that illustrates the chemical structures of 1st, 2nd and 3rd generation tetracycline-class antibiotics (revised manuscript, lines 73-75, 433-437).*

6. There were few grammatical errors in the manuscript. The authors should check them thoroughly.

Response: Every effort was made to check and correct all grammatical errors.

7. The authors should also discuss the plausible reasons for differential activity of these drugs against various clinical strains. Is there any correlation between sequencing data and differential susceptibility of these strains.

Response: The differences in anti-M. abscessus activities exhibited by the four 3rd generation tetracycline-class antibiotics could be attributed in part to variations in chemical structure (revised manuscript, Figure 1). A comparison of the sequencing data derived from all the isolates failed to reveal any obvious genetic differences that could account for differences in drug susceptibility (revised manuscript, lines 238-245).

April 17, 2023

Prof. Haiqing Chu
Shanghai Pulmonary Hospital, Tongji University School of Medicine
No. 507 Zhengmin Road
Shanghai
China

Re: Spectrum00718-23R1 (Omadacycline, Eravacycline and Tigecycline Express Anti-*Mycobacterium abscessus* Activity *In vitro*)

Dear Prof. Haiqing Chu:

Your manuscript has been accepted, and I am forwarding it to the ASM Journals Department for publication. You will be notified when your proofs are ready to be viewed.

Sincerely,

Gyanu Lamichhane
Editor, Microbiology Spectrum
